# The Role of the ALDH Family in Predicting Prognosis and Therapy Response in Pancreatic Cancer

**DOI:** 10.3390/biomedicines13082018

**Published:** 2025-08-19

**Authors:** Xing Wu, Bolin Zhang, Yijun Chen, Bogusz Trojanowicz, Yoshiaki Sunami, Jörg Kleeff

**Affiliations:** 1Department of Visceral, Vascular and Endocrine Surgery, Martin-Luther-University Halle-Wittenberg, University Medical Center Halle, 06120 Halle (Saale), Germany; 2Department of Thyroid Surgery, The Affiliated Hospital of Putian University, Putian 351100, China

**Keywords:** pancreatic cancer, ALDH family, biomarkers, chemoresistance, immune infiltration

## Abstract

**Background**: Pancreatic cancer ranks as the fourth leading cause of cancer-related deaths in the USA. The human aldehyde dehydrogenase (ALDH) family comprises 19 functional members and has been implicated in prognosis and therapy resistance. However, it remains unclear which specific ALDHs are associated with adverse prognoses in pancreatic cancer. **Methods**: We obtained transcriptomic and clinical data for pancreatic adenocarcinoma (PAAD) from the TCGA, corresponding mutational data, and normal pancreatic tissue transcriptomic data from GTEx. Prognostic analysis was carried out using Kaplan–Meier analysis. KEGG and GO analyses were used for biological signaling pathways, and ESTIMATE algorithms were used for tumor microenvironment (TME) assessment. CIBERSORT algorithm, immune infiltration analysis, and OncoPredict algorithms were employed for predicting chemotherapy sensitivity. **Results**: Our study identified four of the 19 ALDH genes (*ALDH1L1*, *ALDH3A1*, *ALDH3B1*, *ALDH5A1*) that were significantly associated with pancreatic cancer prognosis. High expression of *ALDH1L1*, *ALDH3A1*, and *ALDH3B1* was associated with shorter overall survival, while *ALDH5A1* expression was associated with longer overall survival of pancreatic cancer patients. Clinicopathological analysis revealed a significant association with KRAS mutational status and *ALDH3A1* expression. Immune correlation analysis indicated that high expression of *ALDH3A1* and *ALDH3B1* was associated with lower expression of CD8^+^ T cell-associated gene expression. ESTIMATE analyses further revealed that high expression of *ALDH3A1* and *ALDH3B1* was associated with lower levels of immune cell infiltration. PAAD tumors with low *ALDH3A1* expression were more sensitive to paclitaxel. Immunohistochemical analysis demonstrated high expression of ALDH3A1 in pancreatic cancer cells of human tumor tissues compared to normal pancreatic tissues. **Conclusions**: This study unveils specific ALDH family members relevant for prognosis and chemotherapy response in pancreatic cancer patients. These findings contribute valuable insights into prognostic biomarkers and their potential clinical utility in the treatment of pancreatic adenocarcinoma.

## 1. Introduction

Pancreatic cancer (PC) is currently the third leading cause of cancer-related death in the United States and is projected to become the second leading cause by 2030 [1]. Globally, PC ranks among the top causes of cancer-related mortality, with increasing incidence rates worldwide. Unfortunately, most diagnoses occur at an advanced stage because PC is difficult to diagnose. For patients with resected disease, adjuvant therapy, e.g., with modified 5-fluorouracil (5-FU), irinotecan, oxaliplatin, or gemcitabine is recommended [2,3]. Moreover, advancements in neoadjuvant chemotherapy have contributed significantly to improving the management of pancreatic cancer at an advanced local stage and have significantly improved the long-term outcomes for these patients [4]. However, PC is characterized by resistance to standard therapy, such as chemotherapy and targeted therapy [5]. Therefore, the discovery of novel prognostic and therapeutic candidates remains a priority for patients with PAAD.

Human aldehyde dehydrogenases (ALDHs) are a multigene family with 19 functional members [6]. ALDHs, as members of a broad enzyme superfamily, are frequently overexpressed in cancer and have been linked to patient prognosis. ALDHs are versatile enzymes with multiple catalytic functions, including aldehyde oxidation and ester hydrolysis, while also acting as indirect antioxidants through NAD(P)H generation [7]. The relationship between these proteins and chemotherapy resistance and drug targets is well established [8]. Notably, ALDH1A3 exhibits high expression in pancreatic cancer, impacting prognosis and facilitating metastasis [9,10]. Targeting ALDH3A1 in conjunction with chemotherapy offers a potential strategy to overcome individualized drug resistance in lung cancer [11]. ALDH2 is involved in the pathogenesis and progression of diverse cancers and promotes drug resistance [12]. In pancreatic cancer (PC), elevated ALDH activity is connected with worse prognosis and tumor aggressiveness due to its role in maintaining cancer stem cell (CSC) characteristics and promoting resistance to chemotherapy [13]. In addition, pancreatic cancer cells with high ALDH activity have significantly greater tumor-initiating ability than CD133^+^ or ALDH-low cells, supporting ALDH as a reliable marker for pancreatic cancer stem-like cells [14]. Nonetheless, the prognostic significance of all ALDH family members in pancreatic cancer remains unclear. Here, we performed a comprehensive bioinformatic analysis aimed at identifying ALDHs of clinical relevance and prognostic significance in PAAD. We further delved into investigating the correlation between ALDHs and immune infiltration, along with their interplay with chemotherapy. This comprehensive analysis aimed to enhance our understanding of the mechanisms by which ALDHs influence the TME in PC and how targeting ALDHs can help to overcome patient-specific drug resistance.

## 2. Material and Methods

### 2.1. Data Sources

Transcriptomic and clinical data for pancreatic adenocarcinoma (PAAD) were sourced from The Cancer Genome Atlas (TCGA; https://portal.gdc.cancer.gov/, accessed on 10 August 2025), including corresponding mutational data. Raw counts underwent normalization to TPM, followed by log2(TPM + 1) transformation when required for downstream analysis.

### 2.2. Survival and Multivariate Cox Regression Analysis

Kaplan–Meier analysis was conducted via the survival package (https://cran.r-project.org/web/packages/survival/, accessed on 10 August 2025) to evaluate the prognostic relevance of ALDHs by comparing overall survival between patients with high and low expression levels. Samples were categorized into high and low expression groups using the median expression value of each ALDH gene as the cutoff, and the optimal cutoff was also determined with the “surv_cutpoint” function. Independent prognostic factors were identified through multivariate Cox regression with the “coxph” function.

### 2.3. Expression and Clinical Analysis

GEPIA2 (http://gepia2.cancer-pku.cn, accessed on 10 August 2025) was employed to evaluate the expression of genes (*ALDH1L1*, *ALDH3A1*, *ALDH3B1*, *ALDH5A1*) in TCGA and the Genotype-Tissue Expression (GTEx) datasets [15]. Relative gene expression was further compared between high- and low-expression groups across different pathological stages.

### 2.4. Estimation of Tumor Mutational Burden (TMB)

Somatic mutation data were analyzed and visualized using the “maftools” R package(v2.22.0) [16]. The relative expression levels of *ALDH1L1*, *ALDH3A1*, *ALDH3B1*, and *ALDH5A1* were further compared between high- and low-expression groups in the context of KRAS mutation.

### 2.5. Functional Enrichment Analysis

DEGs between high- and low-expression groups of *ALDH1L1*, *ALDH3A1*, *ALDH3B1*, and *ALDH5A1* were identified using “DESeq2”. Functional enrichment, including GO annotation and KEGG pathway analysis, was conducted on differentially expressed genes (DEGs; |log_2_FC| ≥ 1, padj < 0.05) using the “clusterProfiler” [17].

### 2.6. Estimation of Stromal and Immune Scores and Immune Infiltration Analysis

The ESTIMATE (Estimation of STromal and Immune cells in MAlignant Tumor tissues using Expression data) algorithm was applied to infer stromal and immune components within the PAAD tumor microenvironment [18]. Immune cell composition in PAAD samples was estimated using the CIBERSORT algorithm [19]. Gene–gene expression correlations were analyzed using the TCGA dataset through the GEPIA2.

### 2.7. Chemotherapy Response Prediction

To evaluate potential chemotherapy response in PAAD, IC50 values for each patient were estimated with “oncoPredict” using drug response data from the Genomics of Drug Sensitivity in Cancer (GDSC) database [20,21]. Five commonly used chemotherapeutic drugs were selected and their predicted therapeutic effects compared for *ALDH1L1*, *ALDH3A1*, *ALDH3B1*, *and ALDH5A1*.

### 2.8. scRNA-Seq Analysis and Expression Levels of ALDH3A1 in Pancreatic Cancer Cell Lines

Pancreatic cancer single-cell data analysis was carried out using Tumor Immune Single-cell Hub 2 (TISCH2; http://tisch.comp-genomics.org/, accessed on 10 August 2025). Expression profiles of *ALDH1L1*, *ALDH3A1*, *ALDH3B1*, and *ALDH5A1* were examined based on the PAAD_CRA001160 dataset. Additionally, the Human Protein Atlas (HPA; https://www.proteinatlas.org/, accessed on 10 August 2025) was utilized to evaluate ALDH3A1 expression in pancreatic cancer cell lines.

### 2.9. Cell Culture

Human PAAD cell lines (BxPC-3, Capan-1, AsPC-1, Su.86.86, MIA PaCa-2, PANC-1) were cultured in an incubator at 37 °C and 5% CO_2_. Cell lines BxPC-3, AsPC-1, and Su.86.86 were sustained in RPMI-1640 medium (Sigma-Aldrich, Steinheim, Germany) with 10% fetal bovine serum (FBS) (Sigma-Aldrich, Steinheim, Germany), penicillin (100 U/mL), and streptomycin (100 mg/mL) (Thermofisher, Dreieich, Germany). Capan-1 was maintained in RPMI-1640 medium with 15% FBS, penicillin (100 U/mL), and streptomycin (100 mg/mL). PANC-1 was maintained in high-glucose Dulbecco’s Modified Eagle’s medium (DMEM) (Sigma-Aldrich, Steinheim, Germany) with 10% FBS, penicillin (100 U/mL), and streptomycin (100 mg/mL). MIA PaCa-2 was maintained in high-glucose DMEM with 5% FBS, 5% horse serum, penicillin (100 U/mL), and streptomycin (100 mg/mL).

### 2.10. Human Samples

Human PC samples were obtained from the University Medical Center Halle, Martin-Luther-University Halle-Wittenberg, Germany. The use of human samples was conducted in accordance with the ethical principles outlined in the Declaration of Helsinki and was approved under protocol number 2019-037.

### 2.11. Immunohistochemistry

Serial 4 μm sections of paraffin-embedded tissue underwent deparaffinization and rehydration following previously described methods [22]. Slides underwent antigen retrieval in citrate buffer (pH 6.0) using a microwave oven, and non-specific reactivity was then blocked with 1% BSA in PBS. The sections were incubated with the ALDH3A1 antibody (Santa Cruz Biotechnology, Dallas, TX, USA; sc-376089, 1:300) at 4 °C overnight and then incubated with secondary biotinylated antibody (Dako, Santa Clara, CA, USA; K0675) and system HRP (Dako, K0675). Color-reaction with a Dako DAB^+^ chromogen kit (Dako K3468, 1:300) and counterstaining with hematoxylin were carried out. Staining score (0–3) and the proportion of positive staining were quantified using QuPath v0.5.1, and H-scores were calculated as their product.

### 2.12. qPCR

Total RNA was extracted from PDAC cell lines and isolated according to the manufacturer’s protocol using a direct-zol^TM^ Mini-prep Kit (ZYMO RESEARCH, Freiburg, Germany). qPCR was performed using HOT FIREPol^®^ EvaGreen^®^ qPCR Mix Plus (ROX) (Solis BioDyne, Tartu, Estonia). Sequences for qPCR primers are as follows: ALDH3A1 (human): sense: CTC GTC ATT GGC ACC TGG AAC T, antisense: CTC GCC ATG TTC TCA CTC AGC T; β-Actin (human): sense: AGG CAC CAG GGC GTG AT, antisense: GCC CACA TA GGA ATC CTT CTG AC. Gene expression was normalized against β-actin using the 2^−ΔΔCt^ method.

### 2.13. Western Blot

Proteins were separated by 12% SDS-PAGE and then transferred onto PVDF Blotting Membranes (GE Healtcare Life science, Macclesfield, UK). The membranes were incubated overnight at 4 °C with primary ALDH3A1 antibody (1:500; Santa Cruz Biotechnology, sc-376089), then with the secondary antibody for 1 h at room temperature. Protein expression was normalized to β-actin and quantified using ImageJ (v1.53).

### 2.14. Statistical Analysis

All bioinformatics analyses and visualization were performed using R software v4.2.3 (https://www.r-project.org/, accessed on 10 August 2025), and statistical analyses were performed using Prism v10 (GraphPad, San Diego, CA, USA). The Wilcoxon test was applied to compare two groups.

## 3. Results

### 3.1. Prognostic Value of ALDHs in PAAD

Overall survival (OS) in relation to ALDH expression in PAAD patients was evaluated through Kaplan–Meier analysis utilizing the TCGA dataset. In the overall survival analysis, high expression of *ALDH1L1* (*p* = 0.013), *ALDH3A1* (*p* = 0.019), *ALDH3B1* (*p* = 0.003), and low expression of *ALDH5A1* (*p* = 0.032) were found to be significantly correlated with worse survival (Figure 1A–D and Appendix A). Furthermore, optimal cutoff values were determined using the “surv_cutpoint” function, which validated the results (Figure 1E–H). Multivariate analysis was then performed to validate the prognostic relevance of these genes (Figure 1I). These findings indicate that *ALDH1L1*, *ALDH3A1*, *ALDH3B1*, and *ALDH5A1* may be useful as prognostic biomarkers in PAAD.

### 3.2. Differential Expression of ALDH Genes in PAAD: Focus on ALDH3A1 in Tissues and Cell Lines

Next, the expression of *ALDH1L1*, *ALDH3A1*, *ALDH3B1*, and *ALDH5A1* in pancreatic adenocarcinoma (PAAD) samples was compared with that of matching TCGA and GTEx normal samples using GEPIA2. *ALDH3A1* and *ALDH3B1* were found to be upregulated and *ALDH1L1* downregulated in pancreatic cancer versus normal tissues (Figure 2A–C). However, *ALDH5A1* expression did not differ significantly between pancreatic tumors and matched normal tissues (Figure 2D). Single-cell analysis shows that *ALDH3A1* is mainly expressed in pancreatic cancer cells (Figure 2E,G,H). To further validate the differential expression of ALDH3A1 in pancreatic cancer tissues and normal tissues, immunohistochemical staining was performed on human PAAD samples and normal pancreatic tissue samples (Figure 3E–J). ALDH3A1 expression was substantially higher in human pancreatic tumors than in the normal pancreas by immunohistochemistry (Figure 3K). Moreover, Protein Atlas data analysis revealed elevated ALDH3A1 expression in most PC cell lines (Figure 3A). This was further confirmed by qPCR analysis and Western blot of ALDH3A1 expression in PC cell lines (Figure 3B–D). Immunohistochemical staining and single-cell sequencing consistently revealed high ALDH3A1 expression in pancreatic cancer, with localization predominantly in cancer cells within the tumor microenvironment.

### 3.3. Correlation Between ALDHs Expression and Clinicopathological Parameters in PAAD

KRAS mutations play a critical role in the development and progression of PAAD. To further explore the link between KRAS mutation and *ALDH1L1*, *ALDH3A1*, *ALDH3B1*, and *ALDH5A1* expression in cancer progression. Analysis revealed a significant link between KRAS mutation and elevated *ALDH1L1*, *ALDH3A1*, and *ALDH3B1* expression (Figure 4E–G). TMB, quantified as somatic mutations per megabase, was assessed for differences in mutation rates according to gene expression levels. Mutation frequency was higher in the *ALDH1L1* (86.42%), *ALDH3A1* (96.39%), and *ALDH3B1* (95.24%) above median group than in the below median group (Appendix A). This is consistent with the KRAS mutation analysis, which shows that the above median groups of *ALDH1L1*, *ALDH3A1*, and *ALDH3B1* have a higher mutation rate and KRAS mutation frequency, suggesting that they might be involved in pancreatic tumorigenesis and disease progression. However, *ALDH1L1*, *ALDH3A1*, *ALDH3B1*, and *ALDH5A1* expression did not differ significantly according to age, gender, or tumor stage (Figure 4A–D, Appendix A). The observed correlations with KRAS mutations and elevated TMB levels further support a potential role of *ALDH3A1* in PAAD progression.

### 3.4. Enrichment Analysis on ALDHs

To gain further insight into the underlying biological function of PAAD, enrichment analyses for GO terms and KEGG pathways were executed. The results of the GO analysis indicate that *ALDH1L1* is involved in the modulation of chemical synaptic transmission (Figure 5A), while *ALDH3A1* and *ALDH3B1* are involved in the regulation of membrane potential (Figure 5B,C). *ALDH5A1* is involved in the production of immunoglobulins (Figure 5D). *ALDH1L1* is predominantly expressed in astrocytes, and astrocytes labelled with *ALDH1L1* serve an essential function in indirectly influencing chemical synaptic transmission and plasticity through their regulatory functions in the neurogenic environment [23,24].

The KEGG analysis revealed that *ALDH1L1*, *ALDH3A1*, and *ALDH3B1* were significantly enriched in the neuroactive ligand–receptor interaction and pancreatic secretion pathways (Figure 6A–C). *ALDH5A1* was identified as being enriched in the neuroactive ligand–receptor interaction and insulin secretion pathways (Figure 6D). Functional and enrichment analyses suggest a putative involvement of ALDH genes in PC progression through their roles in neural signaling and secretory regulation within the tumor microenvironment.

### 3.5. Analysis of the Contribution of ALDHs to Drug Resistance

Based on the above, GO analysis results indicate that modulating chemical synaptic transmission can subsequently alter ROS levels, indicating the potential involvement of ALDHs in drug resistance. Next, we evaluated the IC50 of selected compounds in each PAAD sample using the oncoPredict algorithm. PAAD tumors with high *ALDH1L1* expression and low *ALDH3A1* expression were more sensitive to paclitaxel (Figure 7E,J). Moreover, tumors with high *ALDH3B1* expression showed increased sensitivity to irinotecan (Figure 7M). In addition, tumors with high *ALDH3B1* and low *ALDH5A1* expression were more responsive to oxaliplatin. (Figure 7N,S). This finding suggests that ALDH expression levels could be a predictive indicator of chemotherapy response in PAAD cases.

### 3.6. Correlations Between ALDHs Expression of Immune Infiltration

The extent of immune infiltration reflects tumor immune evasion and the patient’s immune responsiveness, serving as a key predictor of prognosis and therapeutic outcomes. The ESTIMATE results demonstrated that tumors with low *ALDH3A1* and *ALDH3B1* expression had higher TME scores than those with high expression (Figure 8A,B). Furthermore, the results of the immune infiltration analysis indicated that high expression of *ALDH3A1* and *ALDH3B1* was associated with a lower expression of CD8^+^ T cell-associated gene expression (Figure 8F,G). In addition, *ALDH3A1* and *ALDH3B1* expression showed a significant inverse correlation with the immune checkpoint molecules *PD-1* (*PDCD1*) and *CTLA-4* (Figure 8I–L).

## 4. Discussion

Pancreatic cancer is characterized by late diagnosis, a poor prognosis, and high therapeutic resistance [25]. ALDHs are crucial detoxifying enzymes that protect cells from oxidative stress by converting toxic aldehydes into less harmful carboxylic acids, thereby playing essential roles in a variety of pathological conditions [26]. Previous research has explored the link between ALDH family members, their expression in cancer tissues, prognosis, and involvement in drug resistance [27,28,29]. Among them, ALDH3 family members can either promote tumor progression by enhancing survival, chemoresistance, and stemness through metabolic reprogramming, or suppress it by inducing reductive stress and inhibiting tumor-supportive microenvironments [30]. ALDH activity is essential for maintaining a subpopulation of drug-resistant cancer cells by mitigating the toxicity of reactive oxygen species (ROS). Further, inhibiting ALDH sensitizes these cells to kinase inhibitors, providing a promising combination therapy strategy [29]. ALDH, particularly ALDH1 family members, drives the maintenance of CSCs and mediates resistance to therapy [31,32]. Consistently, following chemotherapy treatment, the number of cells with high ALDH activity (ALDH^+^ cells) may increase, raising the possibility of resistance development [33]. ALDH family members have a specific role in certain tumors. For example, ALDH1A3 is a critical driver of breast CSC plasticity, metabolic reprogramming, and tumor progression [34]. In intrahepatic cholangiocellular carcinoma, ALDH1A1 overexpression correlates with improved survival, highlighting its potential prognostic value [35]. In contrast, reduced ALDH1L1 expression in hepatocellular carcinoma is associated with poor prognosis [36]. ALDH5A1 is downregulated in ovarian cancer, and its high expression predicts improved outcomes, serving as a favorable biomarker [37]. Moreover, ALDH1A1 and ALDH3A1 exhibit elevated expression in squamous cell carcinoma and adenocarcinoma, and their upregulation may contribute to malignant transformation [38]. In this study comprehensively analyzed ALDHs in PAAD and identified four genes (*ALDH1L1*, *ALDH3A1*, *ALDH3B1*, *ALDH5A1*) as prognostic indicators for pancreatic cancer. Furthermore, we analyzed differences in terms of function, clinical parameters, immune microenvironment, and drug sensitivity.

It is further shown that *ALDH3A1* and *ALDH3B1* are overexpressed in tumor tissues and predict poorer patient outcomes. In line with this, a previous study indicated that ALDH3A1 could be used as a prognostic marker in PC and may be involved in mitochondrial energy metabolism [39]. Furthermore, immunohistochemical analysis demonstrated high expression of ALDH3A1 in human tumor tissues compared to normal pancreatic tissues. Additionally, ALDH3A1 is observed in most PC cell lines. The most frequent mutations of pancreatic cancer are KRAS, TP53, CDKN2A, and SMAD4, with a KRAS mutation rate of more than 90%. The results indicate that ALDH3A1 and ALDH3B1 are significantly upregulated in patients with KRAS mutations compared with those with wild-type KRAS. Consistently, Tumor Mutational Burden (TMB) analysis showed KRAS mutations were higher in tumors with high *ALDH3A1* and *ALDH3B1* expression (76% and 81%, respectively) than in those with low expression (44% and 38%, respectively). KRAS is pivotal in the initiation, progressiogn, and local and distant invasion of pancreatic cancer [40]. Yang et al. showed that ALDH3A1 is a key metabolic marker upregulated in pancreatic adenocarcinoma patients with new-onset diabetes, contributing to tumor progression, immune suppression, and poor prognosis [41]. The results indicate that *ALDH3A1* may contribute to the occurrence and development of PAAD.

The tumor microenvironment (TME) is a complex entity comprising the extracellular matrix (ECM), immune cells, fibroblasts, endothelial cells, and other cell types. During tumor progression, interactions between cancer cells and the TME facilitate initiation, growth, and metastasis [42]. The identification of factors that determine the extent of immune infiltration is significant, as increased immune cell infiltration into tumors can predict response to immune therapies [43]. In our analysis, CD8^+^ T cells also showed higher infiltration in tumors with low ALDH3A1 and ALDH3B1 expression. Cytotoxic CD8^+^ T cells, the principal effectors in cancer immunotherapy, are major killers of neoplastic cells [44]. The above findings suggest that ALDHs may relate to immune infiltration and, consequently, to the tumor microenvironment in PAAD.

ALDHs are pivotal in mediating drug resistance observed in numerous cancers. Gene Ontology (GO) analysis reveals that ALDHs influence chemical synaptic transmission, a process that affects reactive oxygen species (ROS) production due to the metabolic demands of active synapses [45]. These results indicate that ALDHs may influence ROS levels. As chemotherapeutics generate elevated levels of oxidative stress in cancer cells, ALDHs may serve to protect cancer cells against these therapeutic approaches by maintaining reactive oxygen species (ROS) at low levels [46]. Furthermore, by modulating chemical synaptic transmission and subsequently altering ROS levels, ALDHs may promote drug resistance. Croker et al. reported that elevated levels of ALDH enzymes in breast cancer cells contribute to chemotherapy resistance [47]. In melanoma, ALDH1A1 mediates resistance to MAPK/ERK inhibitors by activating PI3K/AKT signaling [48]. ALDH1A1-positive ovarian cancer cells have been shown to contribute to resistance to paclitaxel and topotecan [49]. ALDHs have also been observed to display high activity in CSCs and to function as a biomarker for CSCs [50]. In addition, elevated ALDH activity is associated with resistance of CSCs to chemotherapeutic drugs [51]. For instance, ALDH1A1-positive lung CSCs are enriched in EGFR TKI–resistant tumors and contribute to resistance against gefitinib [52]. Our data showed that pancreatic cancer with low ALDH3A1 expression was predicted to be more sensitive to paclitaxel. Similarly, high ALDH3A1 expression has been linked to chemoresistance in paclitaxel plus gemcitabine-resistant pancreatic cancer cells, possibly by modulating intracellular oxidative stress levels [53]. The results indicate that *ALDH3A1* may contribute to chemoresistance in pancreatic cancer.

## 5. Conclusions

*ALDH1L1*, *ALDH3A1*, *ALDH3B1*, and *ALDH5A1* may serve as potential prognostic markers and predictors of chemotherapy response in pancreatic cancer patients. These findings contribute valuable insights into prognostic biomarkers and their potential clinical utility in the treatment of pancreatic adenocarcinoma.

### Limitations of the Study

Although our prognostic and drug resistance analyses are of some significance, the drug resistance findings require in vitro validation and support by clinical data, as the current analysis relies on public datasets that lack detailed treatment and survival information.

## Figures and Tables

**Figure 1 biomedicines-13-02018-f001:**
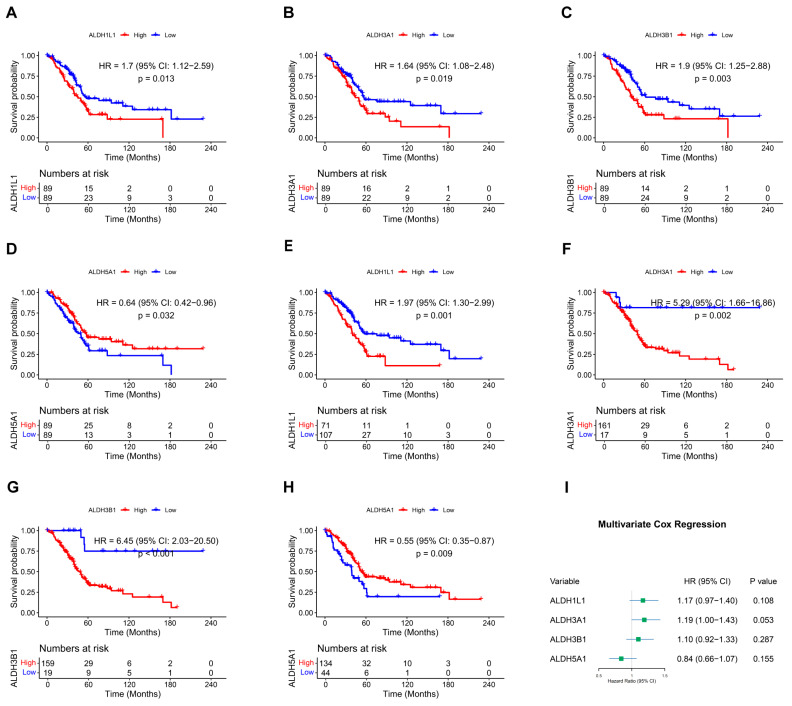
Kaplan–Meier survival curves and multivariate Cox regression analysis. (**A**–**D**) Survival curves of PAAD patients with above or below median expression of *ALDH1L1* (**A**), *ALDH3A1* (**B**), *ALDH3B1* (**C**), *ALDH5A1* (**D**). (**E**–**H**) Survival curves of PAAD patients stratified by optimal cutoff values of *ALDH1L1* (**E**), *ALDH3A1* (**F**), *ALDH3B1* (**G**), and *ALDH5A1* (**H**), determined using the surv_cutpoint function. (**I**) Forest plot showing multivariate Cox regression analysis of *ALDH1L1*, *ALDH3A1*, *ALDH3B1,* and *ALDH5A1* expression in PAAD patients.

**Figure 2 biomedicines-13-02018-f002:**
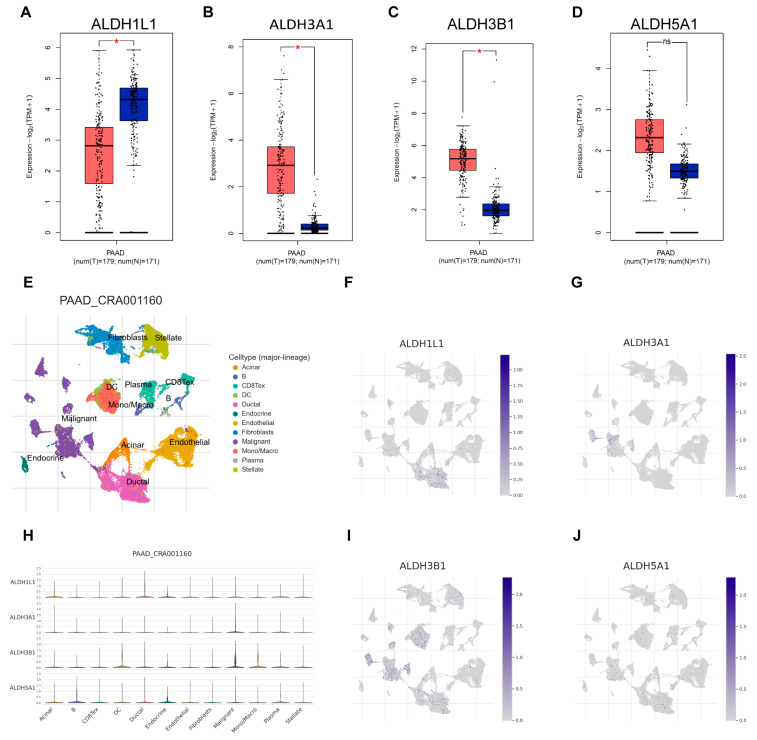
ALDHs expression across cell types in the single-cell transcriptomic dataset PAAD_CRA001160. Comparison of *ALDH1L1* (**A**), *ALDH3A1* (**B**), *ALDH3B1* (**C**), *ALDH5A1* (**D**) expression in pancreatic tumor (red) and normal (blue) tissues. (**E**) UMAP plot displays an integrated cellular map composed of 12 annotated cell types in PAAD_CRA001160. (**F**) UMAP showing ALDH1L1 expression in each cell type. (**G**) UMAP showing ALDH3A1 expression in each cell type. (**H**) VInPlot showing *ALDH1L1*, *ALDH3A1*, *ALDH3B1*, *ALDH5A1* expression in each cell type. (**I**) UMAP showing *ALDH3B1* expression in each cell type. (**J**) UMAP showing *ALDH5A1* expression in each cell type. * *p* < 0.05; ns, not significant.

**Figure 3 biomedicines-13-02018-f003:**
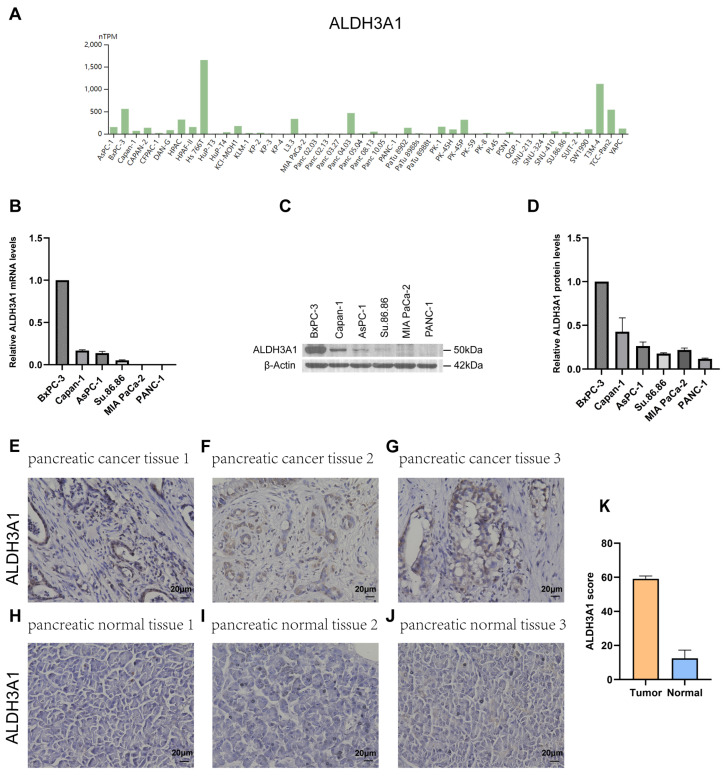
ALDH3A1 expression in pancreatic cancer tissues, normal pancreas, and cancer cell lines. (**A**) Analysis of *ALDH3A1* expression in PC cell lines using HPA. (**B**) *ALDH3A1* expression in PC cells (BxPC-3, Capan-1, AsPC-1, Su.86.86, MIA PaCa-2, PANC-1) was analyzed by qPCR. qPCR was performed in triplicate. Data are presented as mean ± SD. (**C**,**D**) ALDH3A1 expression in PC cells (BxPC-3, Capan-1, AsPC-1, Su.86.86, MIA PaCa-2, PANC-1) was analyzed by Western blot assays. Western blot was performed in triplicate. Data are presented as mean ± SD. (**E**–**G**) Representative immunohistochemistry images showing ALDH3A1 expression in pancreatic cancer tissues. (**H**–**J**) Representative immunohistochemistry images showing ALDH3A1 expression in pancreatic normal tissues. (**K**) Quantification of ALDH3A1 staining scores in pancreatic tumor and normal tissues (n = 3 per group). Bar: 20 μm.

**Figure 4 biomedicines-13-02018-f004:**
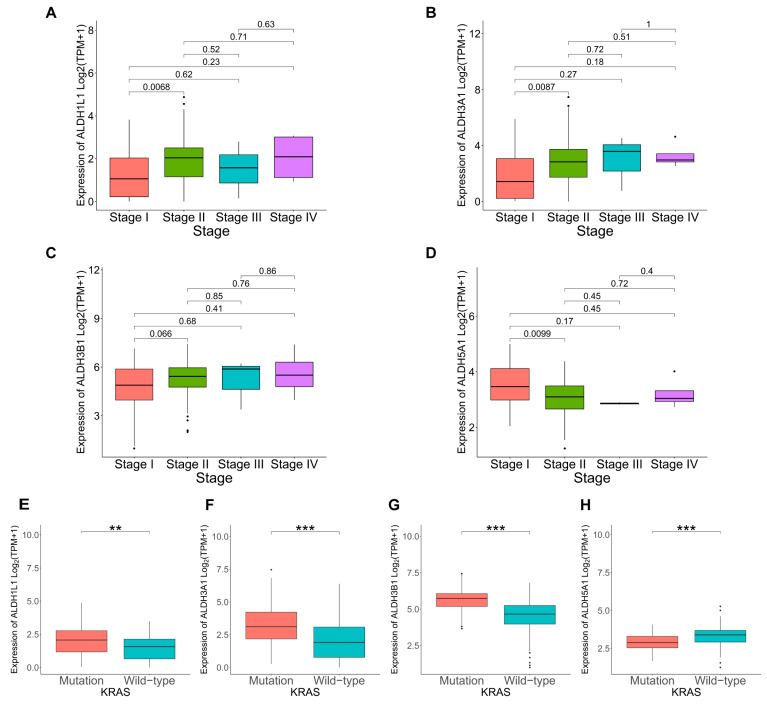
Correlation between ALDHs expression and clinicopathological parameters in PAAD. Correlation of *ALDH1L1* (**A**), *ALDH3A1* (**B**), *ALDH3B1* (**C**), *ALDH5A1* (**D**) expression with cancer stage. Correlation of *ALDH1L1* (**E**), *ALDH3A1* (**F**), *ALDH3B1* (**G**), *ALDH5A1* (**H**) expression with KRAS status. *** *p* < 0.001; ** *p* < 0.01.

**Figure 5 biomedicines-13-02018-f005:**
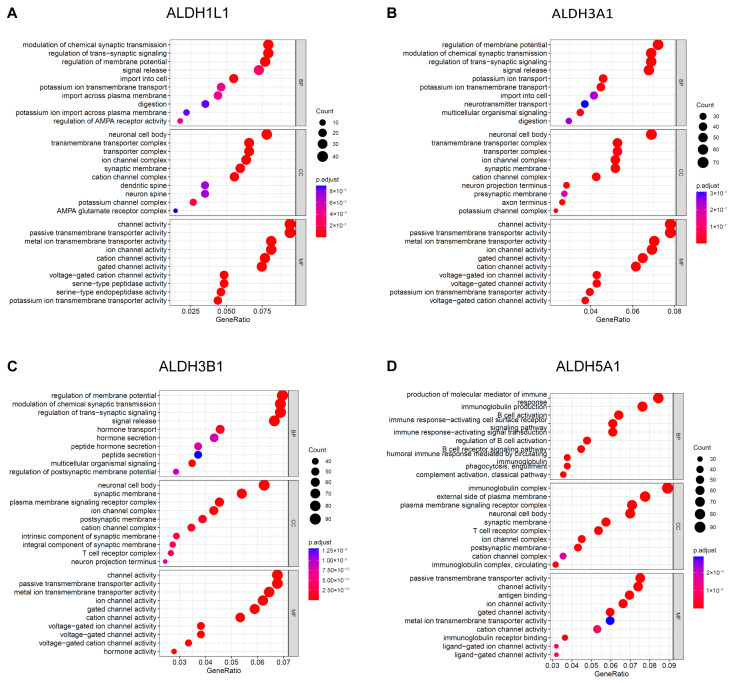
GO functional enrichment analysis of ALDHs in PAAD. Exploring the enrichment of *ALDH1L1* (**A**), *ALDH3A1* (**B**), *ALDH3B1* (**C**), *ALDH5A1* (**D**) expressed above and below median levels in biological functions identified by GO analysis.

**Figure 6 biomedicines-13-02018-f006:**
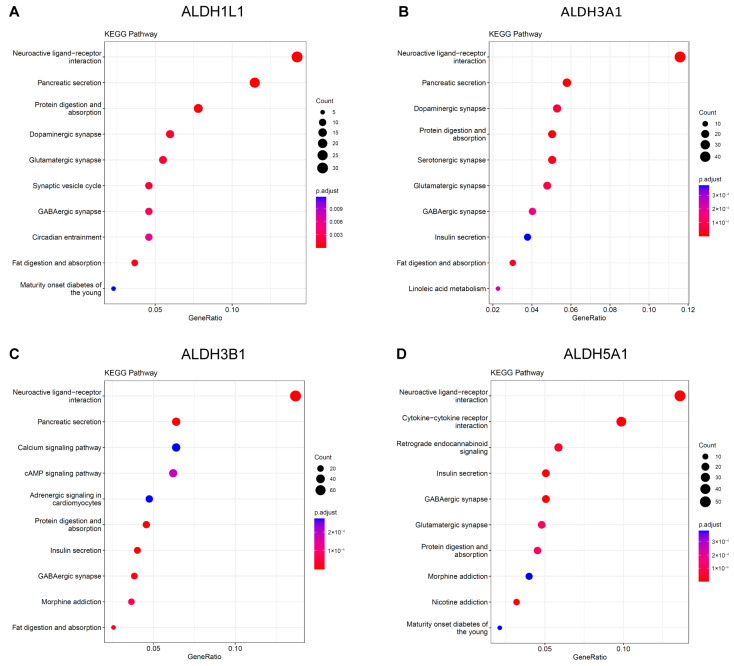
KEGG Pathway analysis of ALDHs in PAAD. Exploring the enrichment of *ALDH1L1* (**A**), *ALDH3A1* (**B**), *ALDH3B1* (**C**), *ALDH5A1* (**D**) expressed above and below median levels by KEGG pathway analysis.

**Figure 7 biomedicines-13-02018-f007:**
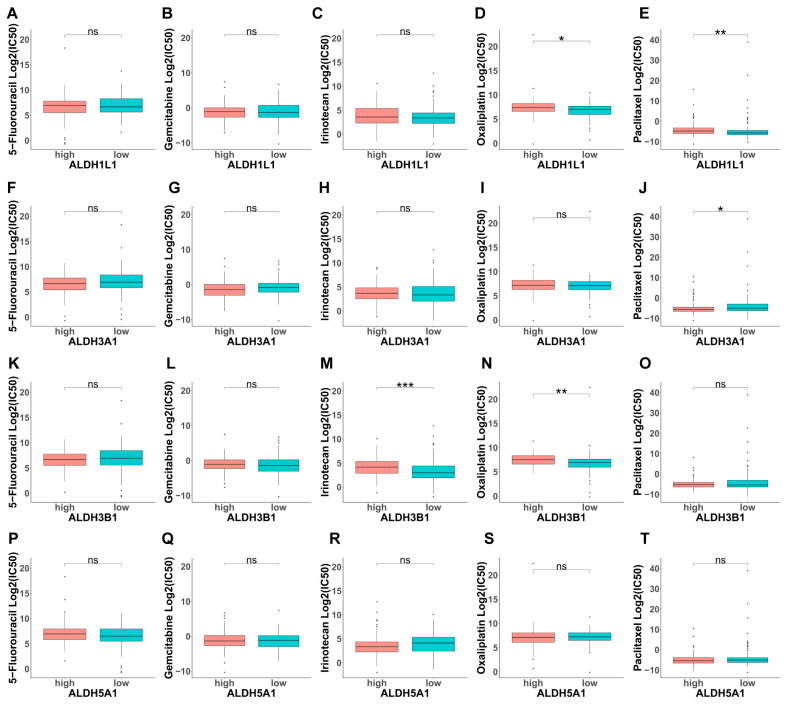
Investigation of drug sensitivity and ALDH expression in PAAD. (**A**–**T**) Comparison of 5-fluorouracil (5-FU), gemcitabine, irinotecan, oxaliplatin, or paclitaxel sensitivity in *ALDH1L1*, *ALDH3A1*, *ALDH3B1*, *ALDH5A1* above and below median expression groups. *** *p* < 0.001; ** *p* < 0.01; * *p* < 0.05; ns, not significant.

**Figure 8 biomedicines-13-02018-f008:**
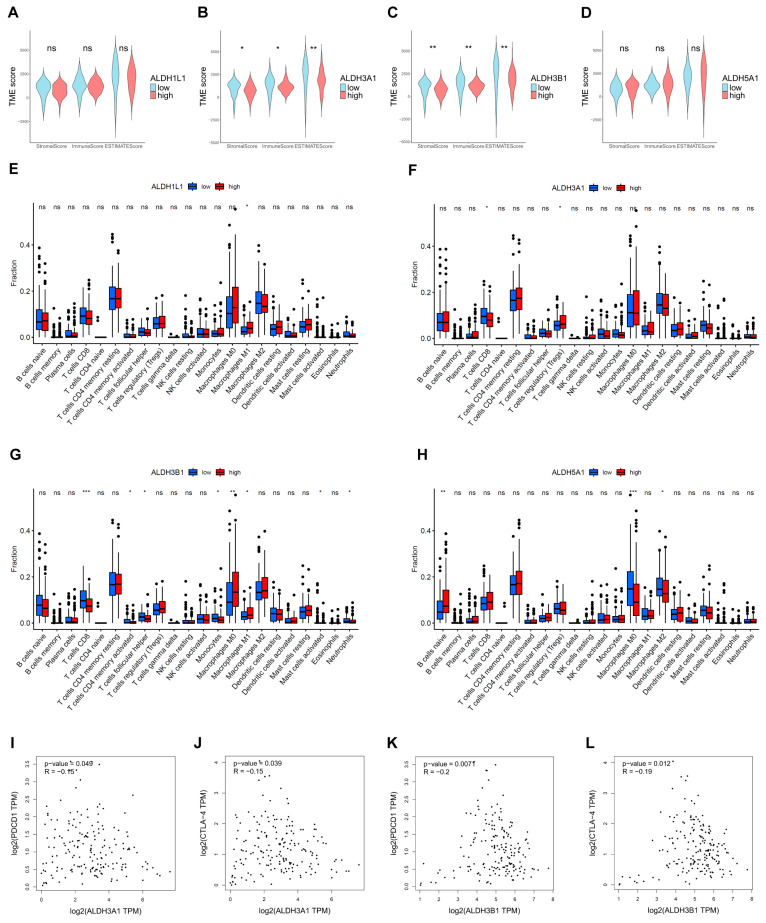
Estimation of TME scores, immune infiltration, and ALDH expression in PAAD. Correlation between *ALDH1L1* (**A**), *ALDH3A1* (**B**), *ALDH3B1* (**C**), *ALDH5A1* (**D**) expression with TME scores by the ESTIMATE algorithm. The relationship between the proportion of immunocytes and the expression of *ALDH1L1* (**E**), *ALDH3A1* (**F**), *ALDH3B1* (**G**), *ALDH5A1* (**H**). (**I**,**J**) Correlation between *ALDH3A1* expression and immune checkpoint molecules *PD-1* (*PDCD1*) and *CTLA-4*. (**K**,**L**) Correlation between ALDH3B1 expression and immune checkpoint molecules *PD-1* (*PDCD1*) and *CTLA-4*. *** *p* < 0.001; ** *p* < 0.01; * *p* < 0.05; ns, not significant.

## Data Availability

The datasets analyzed in this study are available from The Cancer Genome Atlas (TCGA) repository (https://portal.gdc.cancer.gov/).

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
