# Peer review of "The Role of the ALDH Family in Predicting Prognosis and Therapy Response in Pancreatic Cancer"

_biomedicines, 2025, doi:10.3390/biomedicines13082018_

Round 1
Reviewer 1 Report
Comments and Suggestions for Authors
Although the manuscript titled “The Role of the ALDH Family in Predicting Prognosis and Therapy Response in Pancreatic Cancer” presents a compelling, well-investigated, and promising study, however, it cannot be accepted in its current immature form. Substantial revision is required before it is suitable for publication. The following points should be addressed:
- The manuscript states that pancreatic cancer is the fourth leading cause of cancer-related mortality. This information is outdated. The author is advised to incorporate the most recent statistics, which show that pancreatic cancer is currently the third leading cause of cancer mortality in the United States and is projected to become the second leading cause by 2030. Updated references should be cited accordingly:
- American Cancer Society. (2025). Cancer Facts & Figures 2025. Atlanta, GA.
- National Cancer Institute. SEER Cancer Stat Facts: Pancreatic Cancer. Bethesda, MD.
- Pancreatic Cancer Action Network. (2025). Pancreatic Cancer Diagnoses and Mortality Rates Climb; Five‑Year Survival Rate Stalls at 13%. Los Angeles, CA.
- The author needs to address the global burden of pancreatic cancer but not just the US. Add a line on global burden to broaden relevance e.g. “Globally, pancreatic cancer ranks among the top causes of cancer-related mortality, with increasing incidence rates worldwide.”
- The introduction is too brief and lacks essential background information. The author should expand this section to include a broader discussion on the biological significance of ALDH family members. A concise explanation of why ALDH enzymes are particularly relevant in pancreatic cancer including their roles in tumour progression, chemoresistance, and stemness would improve the rationale of the study.
- The material and method section is well written and don’t need any further modification.
- In the result section the title “3.2 Correlation Between ALDH Expression in PAAD and ALDH3A1 Levels in Pancreatic Cancer Tissues, Normal Tissues, and Cell Lines” is a bit confusing because it mentions “ALDH expression” and then narrows to ALDH3A1 within the same phrase, making the scope unclear. Either explain the reason for further investigation of ALDH3A1 or it would be better if the author replaces it with “Differential Expression of ALDH Genes in PAAD: Focus on ALDH3A1 in Tissues and Cell Lines” (Optional).
- In the result section the title “3.2” line 188 The sentence beginning with “To investigate the relationship” should be revised to improve scientific precision. Suggested revision: “To further validate the differential expression of ALDH3A1”.
- In the results section 3.6 the sentence “The sentence: “The results of the ESTIMATE analysis indicated that the TME scores in the ALDH3A1 and ALDH3B1 below median groups were higher than those in the ALDH3A1 and ALDH3B1 above median group. “is confusing and should be rephrased for better clarity.
- In Figure 8A, the gene name label is incorrect. The author mistakenly used “ALDH5A1” instead of “ALDH1L1.” This error should be corrected immediately to maintain scientific accuracy.
- The manuscript contains multiple grammatical and punctuation errors, and several sentences are poorly structured, which may confuse or mislead readers. The entire manuscript should undergo thorough proofreading to ensure clarity and scientific tone.
Multiple sentences are poorly constructed and require substantial revision to enhance clarity, grammatical accuracy, and scientific tone.
Author Response
For research article
|
Response to Reviewer 1 Comments
|
|||||||||||||||
|
1. Summary |
|
|
|||||||||||||
|
Thank you very much for taking the time to review this manuscript. Please find the detailed responses below and the corresponding revisions/corrections highlighted in the re-submitted files. |
|||||||||||||||
|
2. Questions for General Evaluation |
Reviewer’s Evaluation |
Response and Revisions |
|||||||||||||
|
Does the introduction provide sufficient background and include all relevant references? |
Can be improved |
We have revised the Introduction section to provide more comprehensive background information and included additional relevant references to better support the context of our study. |
|||||||||||||
|
Are all the cited references relevant to the research? |
Can be improved |
References were revised to improve relevance. |
|||||||||||||
|
Is the research design appropriate? |
Yes |
|
|||||||||||||
|
Are the methods adequately described? |
Yes |
|
|||||||||||||
|
Are the results clearly presented? |
Yes |
|
|||||||||||||
|
Are the conclusions supported by the results? |
Can be improved |
Conclusions have been revised to better correspond to the results. |
|||||||||||||
|
3. Point-by-point response to Comments and Suggestions for Authors |
|||||||||||||||
|
Comments 1: [The manuscript states that pancreatic cancer is the fourth leading cause of cancer-related mortality. This information is outdated. The author is advised to incorporate the most recent statistics, which show that pancreatic cancer is currently the third leading cause of cancer mortality in the United States and is projected to become the second leading cause by 2030. Updated references should be cited accordingly: American Cancer Society. (2025). Cancer Facts & Figures 2025. Atlanta, GA. National Cancer Institute. SEER Cancer Stat Facts: Pancreatic Cancer. Bethesda, MD. Pancreatic Cancer Action Network. (2025). Pancreatic Cancer Diagnoses and Mortality Rates Climb; Five‑Year Survival Rate Stalls at 13%. Los Angeles, CA.] |
|||||||||||||||
|
Response 1: Thank you for pointing this out. We agree with this comment. Therefore, we have updated the statement to reflect the most recent statistics, indicating that pancreatic cancer is currently the third leading cause of cancer-related mortality in the United States and is projected to become the second leading cause by 2030. The reference list has been updated accordingly. These changes can be found on page 2, paragraph 1, lines 45–46 in the revised manuscript (with changes highlighted in red). “[Pancreatic cancer is currently the third leading cause of cancer-related death in the United States and is projected to become the second leading cause by 2030]”
|
|||||||||||||||
|
Comments 2: [The author needs to address the global burden of pancreatic cancer but not just the US. Add a line on global burden to broaden relevance e.g. “Globally, pancreatic cancer ranks among the top causes of cancer-related mortality, with increasing incidence rates worldwide.] |
|||||||||||||||
|
Response 2: We agree. Thank you for this helpful suggestion. Therefore, we have added a sentence on the global burden of pancreatic cancer to broaden the relevance of the introduction. This revision can be found on page 2, paragraph 1, lines 46–48 of the revised manuscript (marked in red). “[Globally, pancreatic cancer ranks among the top causes of cancer-related mortality, with increasing incidence rates worldwide.]”
|
|||||||||||||||
|
4. Response to Comments on the Quality of English Language |
|||||||||||||||
|
Point 1: Multiple sentences are poorly constructed and require substantial revision to enhance clarity, grammatical accuracy, and scientific tone. |
|||||||||||||||
|
Response 1: Thank you for your constructive feedback. We appreciate your attention to the clarity and quality of the manuscript. These changes aim to enhance the overall readability and ensure more effective communication of our findings. |
|||||||||||||||
|
5. Additional clarifications |
|||||||||||||||
Reviewer 2 Report
Comments and Suggestions for Authors
The manuscript entitled "The role of the ALDH Family in Predicting Prognosis and Therapy Response in Pancreatic Cancer" by Wu et al. presents a comprehensive bioinformatic and partially experimental analysis of the aldehyde dehydrogenase (ALDH) gene family in pancreatic adenocarcinoma (PAAD). The topic is relevant to current clinical practice, as pancreatic cancer remains one of the deadliest malignancies with limited treatment options and well-documented therapeutic resistance. Nevertheless, while the study contains valuable observations, several methodological and interpretative deficiencies must be addressed before the manuscript can be considered for publication.
The Materials and Methods section is generally well organized. It makes appropriate use of established analytical tools. However, important experimental details are lacking. For instance, the number of patient samples used for immunohistochemistry is not provided, nor is any scoring system described (e.g. IRS, H-score), which makes it difficult to assess the robustness of the histological findings. Similarly, the qPCR and Western blot analyses are only briefly mentioned, with no information on replicates or normalization methods. The RNA-seq–based group stratification relies solely on the median expression level, but the study does not discuss the rationale for this cutoff and does not provide a sensitivity analysis using alternative thresholds. Most importantly, multivariate Cox regression was not included in the survival analysis.
In the Results section, the authors identify four ALDH genes (ALDH1L1, ALDH3A1, ALDH3B1, and ALDH5A1) as significantly correlated with overall survival in PAAD. ALDH3A1 and ALDH3B1 are upregulated in tumors and associated with poor prognosis, while ALDH5A1 shows a potential protective effect. These findings are visually well presented, but quantitative details are sparse. Median survival times in high vs. low expression groups are not reported, and hazard ratios with confidence intervals are missing. The reported association between ALDH3A1/3B1 expression and KRAS mutations is interesting and potentially important, but remains observational and lacks mechanistic insight.
The study’s functional analyses (GO and KEGG) suggest enrichment in pathways such as neuroactive ligand-receptor interaction and pancreatic secretion. These findings are relatively non-specific and their relevance to pancreatic tumor biology is not well explained. More insightful would be enrichment in pathways directly related to tumor progression, stemness, metabolism, or immune evasion. The section on immune infiltration is one of the stronger aspects of the study, showing that high expression of ALDH3A1 and ALDH3B1 is associated with lower stromal and immune scores, particularly reduced infiltration of CD8+ T cells. This observation is consistent with an immunosuppressive tumor phenotype, but the study does not include data on checkpoint molecules (e.g., PD-L1), antigen presentation machinery, or cytokines, which would significantly enhance the biological interpretation.
The Discussion section is very long, but at times it is too general and not connected to the data that has been presented. The authors provide a thorough literature overview on ALDHs in various malignancies, but do not sufficiently elaborate on the mechanisms by which ALDH3A1 or ALDH3B1 might promote aggressiveness or therapy resistance specifically in PAAD. Furthermore, the discussion does not take into account other possible explanations, such as the possibility that upregulation of ALDH might be a consequence rather than a cause of aggressive disease. The study's limitations are only mentioned briefly, and more critical reflection is needed — especially about the lack of functional validation and the use of public datasets without matched treatment or survival metadata.
Author Response
For research article
|
Response to Reviewer 2 Comments
|
||||||
|
1. Summary |
|
|
||||
|
Thank you very much for taking the time to review this manuscript. Please find the detailed responses below and the corresponding revisions/corrections highlighted in the re-submitted files. |
||||||
|
2. Questions for General Evaluation |
Reviewer’s Evaluation |
Response and Revisions |
||||
|
Does the introduction provide sufficient background and include all relevant references? |
Yes |
|
||||
|
Are all the cited references relevant to the research? |
Yes |
|
||||
|
Is the research design appropriate? |
Can be improved |
Research design has been reviewed and improved accordingly. |
||||
|
Are the methods adequately described? |
Must be improved |
The Methods section has been revised and expanded based on the reviewers’ comments to provide clearer and more detailed descriptions. |
||||
|
Are the results clearly presented? |
Can be improved |
The Results section has been revised to improve clarity and presentation. |
||||
|
Are the conclusions supported by the results? |
Must be improved |
Based on the reviewers’ comments, additional analyses were conducted and incorporated to strengthen the conclusions. |
||||
|
3. Point-by-point response to Comments and Suggestions for Authors |
||||||
|
Comments 1: [The Materials and Methods section is generally well organized. It makes appropriate use of established analytical tools. However, important experimental details are lacking. For instance, the number of patient samples used for immunohistochemistry is not provided, nor is any scoring system described (e.g. IRS, H-score), which makes it difficult to assess the robustness of the histological findings. Similarly, the qPCR and Western blot analyses are only briefly mentioned, with no information on replicates or normalization methods. The RNA-seq–based group stratification relies solely on the median expression level, but the study does not discuss the rationale for this cutoff and does not provide a sensitivity analysis using alternative thresholds. Most importantly, multivariate Cox regression was not included in the survival analysis.] |
||||||
|
Response 1: We agree. Thank you for your valuable comments regarding the Materials and Methods section and the immunohistochemistry (IHC) analysis. We have now conducted a quantitative assessment of IHC staining using the H-score method. The scoring procedure has been described in detail in the revised Materials and Methods section. Results and the number of samples are shown in Fig. 3K. The corresponding changes can be found on page 4, paragraph 3, lines 159–161 of the revised manuscript (marked in red).
We have revised the manuscript to clarify the qPCR and Western blot procedures. The corresponding changes can be found on page 4, paragraph 4 and 5, lines 169–170 and 176-177 of the revised manuscript (marked in red).
For the RNA-seq–based stratification, in the revised analysis, we addressed this concern by applying the "surv_cutpoint " function to determine optimal cutoff values for group stratification. The corresponding changes can be found on page2, paragraph 3, lines 90–92 of the revised manuscript (marked in red). The corresponding findings are presented in Fig. 1E–H.
Finally, In the revised manuscript, we have included multivariate Cox regression analysis to further validate the independent prognostic value of the variables. The corresponding changes can be found on page3, paragraph 1, lines 92-93 of the revised manuscript (marked in red). The corresponding findings are presented in Fig. 1I. “[Quantification of ALDH3A1 staining scores in pancreatic cancer and normal tissues (n = 3 per group). Staining intensity (scored from 0 to 3) and the percentage of stained area were quantified using QuPath v0.5.1, and H-scores were calculated as the product of intensity and area. Gene expression levels were normalized to β-actin as an internal control using the 2^−ΔΔCt method. qPCR was performed in triplicate. Data are presented as mean ± SD. Protein expression levels were normalized to β-actin as a loading control, and densitometry was performed using ImageJ software. Western blot analysis was performed in triplicate. Data are presented as mean ± SD. Samples were categorized into high and low expression groups using the median expression value of each ALDH gene as the cutoff and the optimal cutoff was also determined using the "surv_cutpoint" function. Multivariate Cox regression analysis was performed using the "coxph" function to identify independent prognostic factors.]”
|
||||||
|
Comments 2: [In the Results section, the authors identify four ALDH genes (ALDH1L1, ALDH3A1, ALDH3B1, and ALDH5A1) as significantly correlated with overall survival in PAAD. ALDH3A1 and ALDH3B1 are upregulated in tumors and associated with poor prognosis, while ALDH5A1 shows a potential protective effect. These findings are visually well presented, but quantitative details are sparse. Median survival times in high vs. low expression groups are not reported, and hazard ratios with confidence intervals are missing. The reported association between ALDH3A1/3B1 expression and KRAS mutations is interesting and potentially important, but remains observational and lacks mechanistic insight.] |
||||||
|
Response 2: We agree. Thank you for your insightful comments. We have added hazard ratios with confidence intervals to the prognostic analysis section to provide more quantitative details; the corresponding changes can be seen in Fig. 1. We agree that the observed association between ALDH3A1/ALDH3B1 expression and KRAS mutations is intriguing and warrants further mechanistic investigation.
|
||||||
|
4. Response to Comments on the Quality of English Language |
||||||
|
Point 1: The English is fine and does not require any improvement. |
||||||
|
Response 1: Thank you very much for your kind words. We appreciate your thorough review and valuable comments. |
||||||
|
5. Additional clarifications |
||||||
|
|
||||||